

# Effect of *Yarrowia lipolytica* yeast biomass with increased kynurenic acid content on selected metabolic indicators in mice

Magdalena Matusiewicz[1], Magdalena Wróbel-Kwiatkowska[2], Tomasz Niemiec[3], Wiesław Świderek[4], Iwona Kosieradzka[3], Aleksandra Rosińska[1], Anna Niwińska[5], Magdalena Rakicka-Pustułka[2], Tomasz Kocki[6], Waldemar Rymowicz[2] and Waldemar A. Turski[6]

[1] Department of Nanobiotechnology, Institute of Biology, Warsaw University of Life Sciences, Warsaw, Poland
[2] Department of Biotechnology and Food Microbiology, Wrocław University of Environmental and Life Sciences, Wrocław, Poland
[3] Department of Animal Nutrition, Institute of Animal Sciences, Warsaw University of Life Sciences, Warsaw, Poland
[4] Department of Animal Genetics and Conservation, Institute of Animal Sciences, Warsaw University of Life Sciences, Warsaw, Poland
[5] Department of Large Animal Diseases and Clinic, Institute of Veterinary Medicine, Warsaw University of Life Sciences, Warsaw, Poland
[6] Department of Experimental and Clinical Pharmacology, Medical University of Lublin, Lublin, Poland

Corresponding author
Magdalena Matusiewicz,
magdalena_matusiewicz@sggw.edu.pl

## ABSTRACT

**Background:** The unconventional yeast species *Yarrowia lipolytica* is a valuable source of protein and many other nutrients. It can be used to produce hydrolytic enzymes and metabolites, including kynurenic acid (KYNA), an endogenous metabolite of tryptophan with a multidirectional effect on the body.
The administration of *Y. lipolytica* with an increased content of KYNA in the diet may have a beneficial effect on metabolism, which was evaluated in a nutritional experiment on mice.
**Methods:** In the dry biomass of *Y. lipolytica* S12 enriched in KYNA (high-KYNA yeast) and low-KYNA (control) yeast, the content of KYNA was determined by high-performance liquid chromatography. Then, proximate and amino acid composition and selected indicators of antioxidant status were compared. The effect of 5% high-KYNA yeast content in the diet on the growth, hematological and biochemical indices of blood and the redox status of the liver was determined in a 7-week experiment on adult male mice from an outbred colony derived from A/St, BALB/c, BN/a and C57BL/6J inbred strains.
**Results:** High-KYNA yeast was characterized by a greater concentration of KYNA than low-KYNA yeast (0.80 ± 0.08 *vs.* 0.29 ± 0.01 g/kg dry matter), lower content of crude protein with a less favorable amino acid composition and minerals, higher level of crude fiber and fat and lower ferric-reducing antioxidant power, concentration of phenols and glutathione. Consumption of the high-KYNA yeast diet did not affect the cumulative body weight gain per cage, cumulative food intake per cage and protein efficiency ratio compared to the control diet. A trend towards lower mean corpuscular volume and hematocrit, higher mean corpuscular hemoglobin concentration and lower serum total protein and globulins was observed, increased

serum total cholesterol and urea were noted. Its ingestion resulted in a trend towards greater ferric-reducing antioxidant power in the liver and did not affect the degree of liver lipid and protein oxidation.

**Conclusions:** The improvement of the quality of *Y. lipolytica* yeast biomass with increased content of KYNA, including its antioxidant potential, would be affected by the preserved level of protein and unchanged amino acid profile. It will be worth investigating the effect of such optimized yeast on model animals, including animals with metabolic diseases.

# INTRODUCTION

*Yarrowia lipolytica* belongs to the unconventional yeast. It was proved that this yeast species can utilize hydrophilic compounds (sugars), as well as hydrophobic (fatty acids, lipids, *etc.*) and use them as a source of carbon and energy (*Hassanshahian, Tebyanian & Cappello, 2012*). These properties result from the presence of hydrolytic enzymes, *e.g.*, hexokinases, lipases, *etc.* and influence the biotechnological potential of *Y. lipolytica*. Thus, the yeast can be applied for the production of hydrolytic enzymes (*Brígida et al., 2014*) and many valuable metabolites, *i.e.*, mannitol (*Tomaszewska, Rywińska & Gładkowski, 2012*), erythritol and citric acid (*Tomaszewska et al., 2014*), arabitol (*Rakicka-Pustułka et al., 2021*) and kynurenic acid (KYNA) (*Wróbel-Kwiatkowska et al., 2020a*, *2020b*).

Furthermore, *Y. lipolytica* has Generally Recognized as Safe (GRAS) status, noted by the US Food and Drug Administration (FDA) and it was approved by EFSA (*European Food Safety Authority, 2019*) as a safe novel food (EU regulation 2015/2283). It was also stated that *Y. lipolytica* biomass can be used as a diet supplement and for animal feeding, as the single-cell protein (SCP) source (*Jach & Malm, 2022*). The reason for this is the fact, that *Y. lipolytica* biomass remains the valuable source of protein, essential amino acids (EAA), minerals, unsaturated fatty acids including essential fatty acids, vitamins (vitamin $B_{12}$ among them) and other compounds (*Jach & Malm, 2022*; *Czech et al., 2016*).

KYNA is an endogenous tryptophan (Trp) metabolite, which possesses valuable properties: antioxidant, anti-inflammatory, anti-virus properties, protective for the nervous system (*Han et al., 2010*) and immune system (*Wirthgen et al., 2018*). KYNA is an agonist of G protein-coupled receptors 35 (GPR35), whose presence is particularly high in the gastrointestinal tract mucosa and immune cells (*Paluszkiewicz et al., 2009*; *Turska et al., 2022*). This compound is easily absorbed from food, is not metabolized, and is mainly excreted in the urine. The positive effect of KYNA on the gastrointestinal tract is known, including its hepatoprotective effect (*Marciniak et al., 2018*; *Pyun et al., 2021*). Despite many valuable properties, natural sources contain small amounts of KYNA and the problem is to introduce it to use on a large scale (*Turska et al., 2022*).

Considering the above, dietary administration of *Y. lipolytica* with increased KYNA content may have a beneficial effect on metabolism, as evaluated in a nutritional experiment on mice.

The objective of the research was to produce the biomass of *Y. lipolytica* S12 enriched with KYNA (*Wróbel-Kwiatkowska et al., 2020b*). Then, KYNA content, proximate and amino acid composition and selected antioxidant indicators of *Y. lipolytica* yeast with different KYNA concentrations were compared. The effect of 5% high-KYNA yeast content in the diet on the growth of laboratory mice, hematological and biochemical indicators of their blood, and liver redox status was determined.

# MATERIALS AND METHODS

## Yeast material

### Strain

The *Y. lipolytica* strain S12 used in the current study belongs to the Yeast Culture Collection of the Department of Biotechnology and Food Microbiology (Wroclaw University of Environmental and Life Sciences, Wrocław, Poland); it was stored on agar slants (4 °C).

### Media

The composition of the slant culture medium for *Y. lipolytica* S12 and the growth medium for seed culture was the same as in *Wróbel-Kwiatkowska et al. (2020a)*. The pH of the second medium was adjusted to the same value as before and in the same manner. The production medium for *Y. lipolytica* S12 (high-KYNA yeast) growth in the bioreactor conditions also had the same composition as before, the medium in the second bioreactor contained 200 mg/L Trp. The control cultures (low-KYNA yeast) were grown without Trp. The pH of the medium was adjusted to 5.3. All media were autoclaved.

### Batch culture

The seed culture was prepared as in *Wróbel-Kwiatkowska et al. (2020a)*, for 48 h. It was then transferred with 10% inoculums (v/v) into a 5-L stirred-tank bioreactor (BIOSTAT B-PLUS; Sartorius, Göttingen, Germany), containing 4 L of a production medium. After 48 h, the culture was transferred into 90-L stirred-tank bioreactor (MPP; New Brunswick Scientific Co., Inc., Edison, NJ, USA), containing 40 L of production medium. All the bioreactor cultures were carried out at 29 °C, for 168 h, the aeration and stirring rates were set at 0.6 v/v/m and 350 rpm. The pH of 5.4 was maintained automatically, by addition of a 20% NaOH or 20% HCl solutions.

### Drying

After centrifugation (4,000 rpm, 5 min), the yeast cells were dried, in a dryer at 50 °C, in the dark, to constant weight. After drying, the yeast biomass was milled and packed in plastic bags and stored at room temperature, in the dark.

Selected parameters of the composition used to evaluate the nutritional and dietetic value of high-KYNA yeast were related to the analogous parameters of low-KYNA yeast, both types of yeast were prepared in the same way for the analyzes.

### Determination of KYNA in yeast biomass

KYNA was isolated and determined by HPLC (high-performance liquid chromatography) method. Dried yeast biomass was mixed with distilled water (1:10, w/v) and centrifuged (5,000 rpm, 10 min). Samples were deproteinated with 2 M perchloric acid, then vortexed, kept at 4 °C, for 10 min and centrifuged at 14,000 × $g$, for 30 min, at 4 °C. The supernatants were applied to a cation-exchange resin (Sigma-Aldrich, St. Louis, MO, USA) prewashed with 0.1 N HCl. The columns were then washed with water and the relevant fraction collected for further analysis. The isolated KYNA was analyzed by a HPLC system (UltiMate 3000 analytical systems; Thermo Fisher Scientific, Waltham, MA, USA). The prepared samples were separated on an analytical column (HC-C18(2); 250 × 4.6 mm inner diameter; 5 µm particle size; Agilent Technologies, Inc, Santa Clara, CA, USA) and quantified fluorometrically (344 nm excitation with 398 nm emission detection). The mobile phase was composed of 20 mmol/L NaAc, 3 mmol/L ZnAc$_2$ and 7% acetonitrile. The flow rate was equal to 1.0 mL/min. Chromeleon 7.2 software (Thermo Fisher Scientific, Waltham, MA, USA) was used to control the HPLC systems and record the chromatographic data. KYNA purchased from Sigma-Aldrich (St. Louis, MO, USA) served as the standard. The analysis was performed in three replicates ($n = 3$).

### Determination of the proximate composition of yeast biomass

The obtained dry yeast biomass was analyzed for contents of crude protein, fat, fiber ($n = 2$) and crude ash ($n = 3$), according to the Association of Official Analytical Chemists (*Association of Official Agricultural Chemists, 2011*; *Wróbel-Kwiatkowska et al., 2020b*). The metabolizable energy was calculated according to the original Atwater equation (*Bielohuby et al., 2010*).

### Analysis of amino acids in yeast biomass

The profile of amino acids was assessed in obtained dry yeast biomass by the method described in *Wróbel-Kwiatkowska et al. (2020b)*. For Trp determination, the samples, after alkaline hydrolysis with lithium hydroxide (110 °C, 16 h) and 4-dimethylaminobenzaldehyde, were tested with colorimetric method, at 590 nm (*Landry & Delhaye, 1992*). $n = 2$.

### Determination of yeast biomass antioxidant indicators

Before each analysis, water extracts from dry yeast biomass, in the optimal concentration, were prepared, as in *Matusiewicz et al. (2022)*.

Ferric-reducing antioxidant power was determined in the extracts according to the modified Oyaizu method (*Matusiewicz et al., 2019*, *2022*), $n = 6$.

Total phenol concentration was determined using the Folin-Ciocalteu method (*Matusiewicz et al., 2019*, *2022*), $n = 6$.

The GSH content was evaluated by the Ellman method (*Matusiewicz et al., 2019*, *2022*), $n = 6$.

## Nutritional experiment and preparation of animal material

A 7-week nutritional research was carried out on healthy male mice from an outbred colony of the C selection line derived from A/St, BALB/c, BN/a and C57BL/6J inbred strains (Department of Animal Genetics and Conservation, Institute of Animal Sciences, Warsaw University of Life Sciences, Warsaw, Poland), in the rodent house of the above Department. The experimental procedures were approved by the local ethics committee (Resolution No. WAW 2/105/2022 of the II Local Ethics Committee on Animal Experiments, located in Warsaw University of Life Sciences, Warsaw, Poland). Forty-eight three-month-old mice with completed somatic growth, having an average initial body weight of 40.5 g, were randomly assigned to four groups ($n = 12$, the need to obtain appropriate volumes of whole blood from six mice and blood serum from another six animals determined the number of animals in the group). Each group of mice was divided into four plastic growth cages ($n = 4$, there were three mice in one cage), with controlled environmental conditions (21 °C, 12/12 h, 40% humidity). WŚ was aware of the group allocation at the all stages of the experiment. The animals were fed *ad libitum* one of four semi-synthetic, isoprotein and isoenergetic diets: (1) control diet; a diet supplemented with (2) commercial KYNA (0.040 g/kg; Sigma-Aldrich, St. Louis, MO, USA); (3) 5% dry low-KYNA yeast; or (4) 5% dry high-KYNA yeast. The diets were prepared by the Zoolab (Sędziszów, Poland) and covered mice nutritional requirements (*NRC, 1996*; *Reeves, 1997*). The nutritional value of the diets was determined according to AOAC procedures (*Association of Official Agricultural Chemists, 2011*), $n = 2$ (crude protein, fat and fiber), $n = 3$ (crude ash, dry matter). The metabolizable energy was calculated as in the section Determination of the proximate composition of yeast biomass. The animals had free access to water. The environment of the animals was not enriched so that the potential ingestion of additional elements did not influence the results of the nutritional experiment. During the 7 weeks of the experiment, the body weight of individually marked mice was monitored weekly, in the same order, starting with group 1 animals (cages 1–4) and ending with group 4 animals (cages 1–4). Body weight per cage was calculated and food intake per cage was also measured weekly. Cumulative body weight gain per cage (g) and cumulative food intake per cage (g/100 g body weight) were calculated. On the 49[th] day of the experiment, in order to collect biological material for analyzes, after an overnight fast, animals were weighed and euthanized by inhalation of isoflurane anesthetic (Aerrane; Baxter, Deerfield, IL, USA) in a desiccator. This method was chosen because it is accepted for euthanasia of laboratory rodents and isoflurane is the preferred inhaled anesthetic for euthanasia of small animals (*Underwood & Anthony, 2020*). Single mice from groups 1–4 (cages 1–4) were delivered to the dissection room one by one and subjected to euthanasia. Mice blood was immediately collected from the heart, from six animals in each group—into EDTA anticoagulant tubes (intended for determination of hematological blood parameters) and from another six mice—into clot activator tubes. After clotting, blood was centrifuged ($2,000 \times g$, 10 min, 4 °C), blood serum, intended for the determination of biochemical parameters, was separated and stored at −25 °C. The liver (left lateral lobe) was dissected

and rinsed in cold PBS (phosphate buffered saline), pH 7.4, frozen in liquid nitrogen and stored ($-80$ °C).

## Analysis of amino acids in experimental diets

The amino acid profile in diets ($n = 2$) was determined as in the section "Analysis of amino acids in yeast biomass".

## Protein efficiency ratio

PER, an index of dietary protein quality, was calculated as the ratio of the cumulative body weight gain per cage (g) to the cumulative amount of crude protein consumed per cage (g), $n = 4$.

## Determination of hematological and biochemical blood parameters

Blood hematological indices: red blood cells, hemoglobin, hematocrit, mean corpuscular volume, mean corpuscular hemoglobin concentration, white blood cells and platelets were determined using the Abacus Junior Vet analyzer (Diatron MI Zrt., Budapest, Hungary). Differential leukocyte counts were performed. Blood serum biochemical indices: total protein, albumins, total cholesterol (TC), high-density lipoproteins (HDL), triacylglycerols (TG), glucose, aspartate aminotransferase (AST), alanine aminotransferase (ALT), lactate dehydrogenase (LDH), gamma-glutamyl transpeptidase (GGTP), urea and creatinine were assayed with a Miura One analyzer (I.S.E. S.r.l., Guidonia Montecelio, Italy). The analyzes were performed in the Veterinary Diagnostic Laboratory, Small Animal Clinic, Warsaw University of Life Sciences, Warsaw, Poland. The amount of globulins was determined by subtracting the amount of albumins from the total protein amount. The albumin/globulin ratio, the TC/HDL ratio and the urea/creatinine ratio = (urea $\times$ 1,000)/creatinine were calculated.

## Determination of redox state indicators in mice livers

Extracts from the livers of mice fed four diets were prepared, $n = 6$. Livers (about 100 mg) were homogenized in cold deionized water (1 mL), using a TissueLyser LT bead homogenizer (Qiagen, Hilden, Germany) with a cooled adapter (50 l/s, 7 min). The homogenates were then centrifuged twice at 5,000 rpm (10 min, 4 °C) and once at $14,000 \times$ g (10 min, 4 °C). The obtained supernatants (extracts) were vortexed and transferred to three Eppendorf tubes intended for further analyzes, frozen in liquid nitrogen and stored at $-80$ °C.

Ferric-reducing antioxidant power in appropriately diluted liver extracts was determined as in the section "Determination of yeast biomass antioxidant indicators".

Thiobarbituric acid reactive substances (TBARS), products of lipid peroxidation, in the extracts were evaluated according to the procedure of Uchiyama & Mihara (*Matusiewicz et al., 2018*, *2022*).

The content of protein carbonyls, metal-catalyzed oxidation products, in diluted liver extracts was assayed according to the modified (*Levine et al., 1990*) method. A total of 250 µL of each liver extract was transferred to Eppendorf tubes in two replicates (test sample and blank sample), then, for the precipitation of nucleic acids, 27.8 µL of 10%

streptomycin sulfate in 50 mM HEPES buffer, pH 7.2 was added. The samples were vortexed, incubated (15 min) and centrifuged (11,000 × g, 10 min). A total of 200 μL of the supernatants were transferred to new tubes and, for protein concentration, 200 μL of 20% TCA was added, the samples were vortexed, centrifuged (11,000 × g, 10 min) and the supernatants were thoroughly discarded. Then, 500 μL of 10 mM 2,4-dinitrophenylhydrazine in 2 M HCl was added to test samples and 500 μL of 2 M HCl to blank samples. The samples were vortexed and sonicated (Vibra-Cell™ Ultrasonic Liquid Processor; Sonics & Materials, Inc., Newton, CT, USA; 500 W, 20 kHz, amplitude 20% for 2 min and amplitude 100% for 12 min) until the pellets were dissolved. After 1 h of incubation (RT), 500 μL of 20% TCA was added to the samples, they were vortexed, centrifuged (11,000 × g, 3 min) and the supernatants were thoroughly discarded. To get rid of free reagent, the pellets were washed twice with 1 mL of ethanol-ethyl acetate (1:1). Prior to each centrifugation (11,000 × g, 3 min), samples were left for 10 min and the supernatants were carefully removed each time. The proteins were thoroughly dissolved in 0.6 mL of 6 M guanidine hydrochloride with 20 mM potassium phosphate, adjusted to pH = 2.3, then the samples were vortexed and incubated in a water bath (37 °C, 15 min). The insoluble material was removed by centrifugation (11,000 × g, 3 min). A total of 200 μL of the supernatants were placed in a 96-well plate and the absorbance was measured at 375 nm (Infinite M200 microplate reader; Tecan, Männedorf, Switzerland).

The absorbance of the blank samples was subtracted from the absorbance of the test samples. The carbonyl content in the wells (C) was calculated using the following formula: C (nmole/well) = ($Absorbance_{375}$/6.364) × 200. The carbonyl content in the livers was finally expressed in μmol/g.

## Statistical analysis

The experimental results are presented as the mean ± the standard error of the mean (SEM) and in the case of box and whiskers plots, boxes enclose the interquartile range, whiskers—min to max values, horizontal lines—the median, "+"—the mean. The results of the elements of yeast chemical composition and its antioxidant status were subjected to an unpaired Student's t-test. The results of the amino acid composition of diets and research in mice were submitted to a one-way analysis of variance (ANOVA) and the means were compared using the Tukey's *post-hoc* test. The differences between the means were statistically significant when $p < 0.05$. Prism 5 (GraphPad Software Inc., San Diego, CA, USA) and Statgraphics Centurion (StatPoint Technologies, Inc., Warrenton, VA, USA) softwares were employed.

## RESULTS

### KYNA content in yeast biomass and its proximate composition

The high-KYNA yeast was characterized by a significantly higher concentration of KYNA, significantly lower level of crude protein and crude ash and lower metabolizable energy than low-KYNA yeast (Table 1). It contained significantly more crude fat and more than twice the crude fiber.

**Table 1 The content of kynurenic acid (KYNA) in yeast biomass and its proximate composition.**

| Compound (g/kg) | Low-KYNA yeast | High-KYNA yeast | $p$ |
|---|---|---|---|
| KYNA | 0.29 ± 0.01 | 0.80 ± 0.08 | 0.003 |
| Crude protein | 446.4 ± 7.7 | 303.4 ± 0.1 | 0.003 |
| Crude fat | 14.4 ± 1.0 | 19.6 ± 0.7 | 0.050 |
| Crude fiber | 72.4 ± 1.3 | 160.6 ± 0.5 | <0.001 |
| Crude ash | 54.4 ± 0.2 | 25.2 ± 0.0 | <0.001 |
| Metabolizable energy (kcal/100 g) | 355.9 | 334.4 | – |

**Note:**
Data are expressed as mean ± standard error of the mean. Statistically significant effect: values of one compound are statistically significantly different when $p < 0.05$. $n = 3$ (KYNA, crude ash), $n = 2$ (crude protein, fat and fiber).

**Table 2 Amino acid composition of yeast biomass (g/kg crude protein).**

| Amino acids | Low-KYNA yeast | High-KYNA yeast | $p$ |
|---|---|---|---|
| **Essential amino acids (EAA)*** | | | |
| Lysine | 73.65 ± 2.45 | 50.57 ± 0.01 | 0.011 |
| Leucine | 63.44 ± 1.47 | 33.51 ± 1.62 | 0.005 |
| Threonine | 54.12 ± 4.73 | 63.80 ± 0.86 | 0.181 |
| Valine | 52.53 ± 0.72 | 39.85 ± 0.67 | 0.006 |
| Isoleucine | 41.87 ± 0.99 | 28.96 ± 0.88 | 0.010 |
| Phenylalanine | 41.21 ± 1.16 | 26.41 ± 0.80 | 0.009 |
| Methionine | 18.77 ± 0.34 | 10.18 ± 0.34 | 0.003 |
| Tryptophan | 11.24 ± 0.18 | 9.50 ± 0.06 | 0.011 |
| **Semi-essential amino acids (SEAA)*** | | | |
| Arginine | 41.72 ± 1.48 | 22.38 ± 0.15 | 0.006 |
| Histidine | 25.74 ± 0.99 | 18.30 ± 0.39 | 0.020 |
| **Non-essential amino acids (NEAA)*** | | | |
| Glutamic acid | 154.94 ± 5.50 | 94.62 ± 1.36 | 0.009 |
| Aspartic acid | 101.75 ± 2.27 | 78.88 ± 1.03 | 0.012 |
| Alanine | 84.02 ± 1.99 | 65.02 ± 0.87 | 0.013 |
| Serine | 59.43 ± 1.26 | 67.84 ± 0.45 | 0.024 |
| Glycine | 56.46 ± 1.91 | 48.24 ± 0.65 | 0.055 |
| Tyrosine | 53.52 ± 1.96 | 21.35 ± 0.86 | 0.004 |
| Proline | 46.58 ± 2.15 | 51.50 ± 0.48 | 0.154 |
| Cysteine | 12.75 ± 0.32 | 10.76 ± 0.28 | 0.042 |
| **Amino acid groups** | | | |
| Total amino acids | 993.69 ± 31.47 | 741.63 ± 6.83 | 0.016 |
| Essential amino acids | 356.81 ± 11.66 | 262.77 ± 2.82 | 0.016 |
| Semi-essential amino acids | 67.45 ± 2.47 | 40.68 ± 0.23 | 0.008 |
| Non-essential amino acids | 569.44 ± 17.35 | 438.19 ± 4.25 | 0.018 |

**Note:**
Data are expressed as mean ± standard error of the mean. * for humans. Statistically significant effect: values of one parameter are statistically significantly different when $p < 0.05$. $n = 2$. KYNA, kynurenic acid.

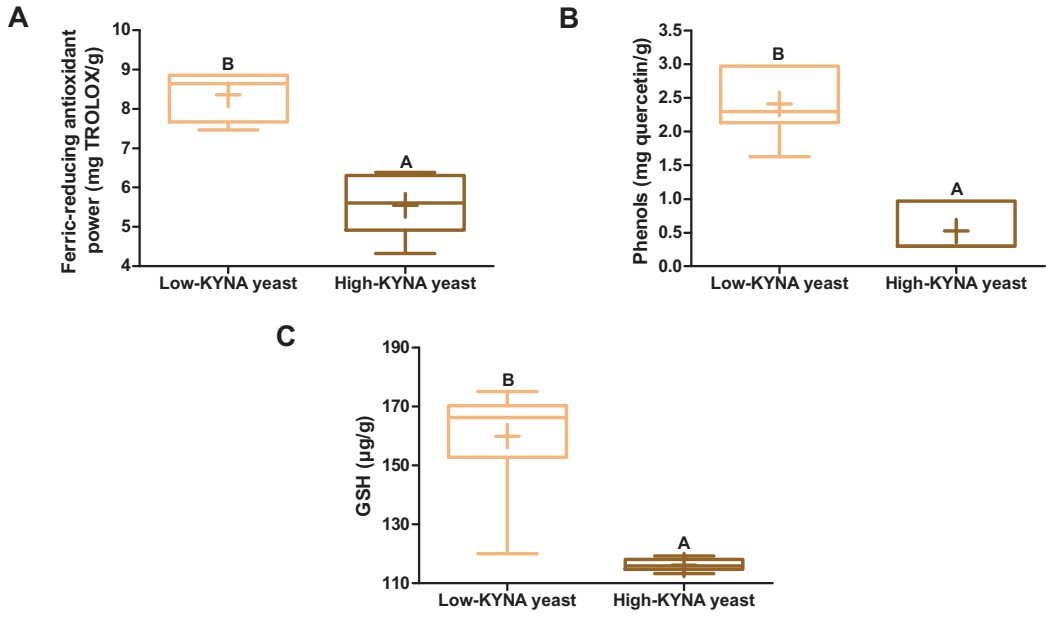

**Figure 1 Box and whiskers plots for antioxidant indicators.** (A) Ferric-reducing antioxidant power and concentration of (B) phenols and (C) GSH (glutathione) in yeast. Boxes enclose the interquartile range, whiskers—min to max values, horizontal lines—the median, "+"—the mean. Statistically significant effect: values of one parameter without common letter (A,B) are statistically significantly different at a significance level of $p < 0.01$. $n = 6$. KYNA, kynurenic acid.

## Analysis of amino acids in yeast biomass

Significantly lower levels of most measured amino acids were found in the high-KYNA yeast crude protein, except for threonine (Thr), serine (Ser), glycine (Gly) and proline (Pro), compared to the low-KYNA yeast protein (Table 2). The concentration of Ser was significantly higher, and the levels of Thr, Gly and Pro were unchanged. The contents of total amino acids (TAA), EAA, semi-essential amino acids (SEAA) and non-essential amino acids (NEAA) in the crude protein were also significantly lower in this yeast.

## Antioxidant indicators in yeast biomass

The high-KYNA yeast was characterized by statistically significantly lower ferric-reducing antioxidant power, the concentration of phenols and GSH than the low-KYNA yeast (Figs. 1A–1C).

## Composition and nutritional value of diets, and analysis of amino acids

The composition and nutritional value (content of crude protein, crude fat, crude fiber, crude ash, dry matter and metabolizable energy) of isoprotein and isoenergetic experimental diets are presented in Table 3. The first diet was a control diet, the second-KYNA diet contained KYNA in the amount of 0.04 g/kg (KYNA concentration equals KYNA content of yeast in high-KYNA yeast diet), the third-low-KYNA yeast diet was a diet comprising 5% low-KYNA yeast, and the last-high-KYNA yeast diet contained 5% high-KYNA yeast.

**Table 3 Composition and nutritional value of diets.**

| Component (g/kg) | Experimental diet | | | |
|---|---|---|---|---|
| | Control | KYNA | Low-KYNA yeast | High-KYNA yeast |
| KYNA | – | 0.04 | – | – |
| Low-KYNA yeast | – | – | 50.00 | – |
| High-KYNA yeast | – | – | – | 50.00 |
| Others | | | | |
| Nutritional value (g/kg) | | | | |
| Crude protein | 122.66 ± 1.86 | 125.10 ± 1.30 | 129.20 ± 0.10 | 129.95 ± 4.55 |
| Crude fat | 33.25 ± 0.45 | 33.15 ± 1.05 | 32.35 ± 1.15 | 31.10 ± 0.80 |
| Crude fiber | 15.64 ± 1.04 | 14.24 ± 1.77 | 9.95 ± 0.05 | 10.85 ± 0.56 |
| Crude ash | 26.05 ± 0.44 | 26.48 ± 0.85 | 29.56 ± 0.42 | 27.18 ± 0.23 |
| Dry matter | 913.12 ± 0.98 | 917.03 ± 0.38 | 919.27 ± 0.68 | 915.01 ± 0.75 |
| Metabolizable energy (kcal/100 g) | 398.9 | 401.3 | 401.3 | 401.3 |

Note:
Data concerning nutritional value are expressed as mean ± standard error of the mean, $n = 2$ (crude protein, fat and fiber), $n = 3$ (crude ash, dry matter). KYNA, kynurenic acid.

**Table 4 Amino acid composition of diets (g/kg crude protein).**

| Amino acids | Experimental diet | | | | $p$ |
|---|---|---|---|---|---|
| | Control | KYNA | Low-KYNA yeast | High-KYNA yeast | |
| **Essential amino acids (EAA)**[*] | | | | | |
| Leucine | 84.84 ± 0.39 | 83.40 ± 3.88 | 76.75 ± 0.12 | 83.71 ± 0.37 | 0.128 |
| Lysine | 82.40 ± 0.53[B] | 78.68 ± 0.41[B] | 71.88 ± 1.48[A] | 77.97 ± 0.33[B] | 0.004 |
| Valine | 57.99 ± 1.65 | 56.99 ± 0.80 | 52.86 ± 0.54 | 55.56 ± 0.42 | 0.072 |
| Phenylalanine | 46.02 ± 0.56[b] | 46.34 ± 0.25[b] | 40.48 ± 1.41[a] | 42.87 ± 0.47[ab] | 0.018 |
| Threonine | 38.67 ± 0.39 | 38.77 ± 0.62 | 39.08 ± 0.68 | 41.16 ± 0.37 | 0.081 |
| Isoleucine | 37.13 ± 0.03 | 36.54 ± 1.34 | 35.28 ± 0.11 | 35.44 ± 0.05 | 0.297 |
| Methionine | 28.77 ± 0.65[A] | 28.65 ± 0.13[A] | 26.54 ± 0.50[A] | 43.21 ± 0.52[B] | <0.001 |
| Tryptophan | 16.07 ± 0.29 | 15.28 ± 0.32 | 15.45 ± 0.26 | 15.08 ± 0.08 | 0.170 |
| **Semi-essential amino acids (SEAA)**[*] | | | | | |
| Histidine | 30.50 ± 0.01[b] | 29.98 ± 0.27[b] | 26.46 ± 0.80[a] | 28.42 ± 0.46[ab] | 0.013 |
| Arginine | 28.09 ± 0.43 | 28.08 ± 0.67 | 27.14 ± 0.42 | 32.66 ± 3.16 | 0.220 |
| **Non-essential amino acids (NEAA)**[*] | | | | | |
| Glutamic acid | 228.77 ± 11.83 | 235.85 ± 0.10 | 210.37 ± 4.85 | 226.68 ± 2.92 | 0.181 |
| Proline | 103.70 ± 3.82[b] | 90.02 ± 4.25[ab] | 85.10 ± 2.30[a] | 97.90 ± 0.78[ab] | 0.045 |
| Aspartic acid | 69.91 ± 0.12 | 68.36 ± 0.42 | 68.39 ± 1.78 | 69.93 ± 0.44 | 0.515 |
| Serine | 52.66 ± 0.21[ab] | 54.00 ± 0.34[ab] | 51.84 ± 0.92[a] | 55.33 ± 0.24[b] | 0.031 |
| Alanine | 25.77 ± 0.37[A] | 27.07 ± 0.03[B] | 33.50 ± 0.03[D] | 28.64 ± 0.05[C] | <0.001 |
| Tyrosine | 23.81 ± 1.09[ab] | 25.65 ± 0.13[b] | 20.93 ± 0.72[a] | 25.32 ± 0.45[b] | 0.026 |
| Glycine | 16.12 ± 0.22[A] | 16.68 ± 0.08[A] | 20.72 ± 0.24[C] | 18.88 ± 0.41[B] | <0.001 |
| Cysteine | 4.45 ± 0.07[A] | 4.44 ± 0.12[A] | 5.48 ± 0.07[B] | 5.85 ± 0.17[B] | 0.002 |

| Amino acids | Experimental diet | | | | p |
|---|---|---|---|---|---|
| Table 4 (continued) | | | | | |
| | Control | KYNA | Low-KYNA yeast | High-KYNA yeast | |
| | Amino acid groups | | | | |
| Total amino acids | 975.66 ± 4.66[B] | 964.75 ± 0.73[B] | 908.21 ± 13.89[A] | 984.56 ± 7.98[B] | 0.010 |
| Essential amino acids | 391.89 ± 2.29[B] | 384.64 ± 5.04[B] | 358.30 ± 3.35[A] | 394.98 ± 0.72[B] | 0.004 |
| Semi-essential amino acids | 58.59 ± 0.42 | 58.06 ± 0.40 | 53.60 ± 1.22 | 61.07 ± 3.61 | 0.188 |
| Non-essential amino acids | 525.18 ± 7.35 | 522.05 ± 4.71 | 496.31 ± 9.33 | 528.52 ± 3.66 | 0.079 |

Note:
Data are expressed as mean ± standard error of the mean. * for humans. Statistically significant effect: values of one parameter without common superscript are statistically significantly different (a,b—at a significance level of $p < 0.05$; A,B,C,D—at a significance level of $p < 0.01$). $n = 2$. KYNA, kynurenic acid.

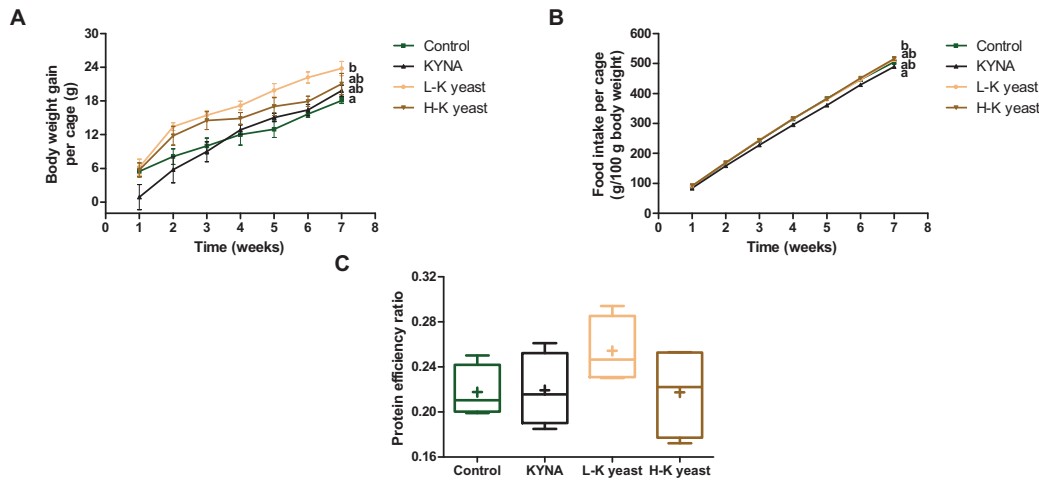

Figure 2 Growth indicators of mice. (A) Cumulative body weight gain per cage (three mice) (g); (B) cumulative food intake per cage (g/100 g body weight) and (C) protein efficiency ratio (PER). Scatter plots—error bars represent standard error of the mean. Box and whiskers plots—boxes enclose the interquartile range, whiskers—min to max values, horizontal lines—the median, „+"—the mean. Statistically significant effect: values of one parameter without common letter (a,b) are statistically significantly different at a significance level of $p < 0.05$. In the case of A and B, the final values were subjected to statistical analysis. $n = 4$. Control, control diet; KYNA, kynurenic acid diet; L-K yeast, low-KYNA yeast diet; H-K yeast, high-KYNA yeast diet.

After balancing the crude protein level of the diets with casein, it turned out that the crude protein of the high-KYNA yeast diet was significantly richer in lysine (Lys), methionine (Met), Ser, tyrosine (Tyr), TAA and EAA, and contained significantly less alanine (Ala) and Gly compared to the crude protein of the low-KYNA yeast diet (Table 4).

### Body weight gain, food intake and protein efficiency ratio
The cumulative body weight gain per cage of mice fed the high-KYNA yeast diet was not significantly different from the cumulative body weight gain per cage of mice fed the control and other diets (Fig. 2A), with no significant differences in cumulative food intake per cage and PER between this group and the control diet group (Figs. 2B, 2C). It is worth

**Table 5  Hematological blood parameters.**

| Parameter | Experimental diet | | | | p |
|---|---|---|---|---|---|
| | Control | KYNA | Low-KYNA yeast | High-KYNA yeast | |
| Red blood cells ($10^6$/mm$^3$) | 8.71 ± 0.38 | 8.64 ± 0.14 | 9.16 ± 0.14 | 8.44 ± 0.63 | 0.612 |
| Hemoglobin (g/dL) | 13.9 ± 1.0 | 13.8 ± 0.1 | 14.3 ± 0.2 | 13.1 ± 0.9 | 0.643 |
| Hematocrit (%) | 46.46 ± 2.20[ab] | 41.00 ± 1.25[ab] | 47.48 ± 0.81[b] | 40.58 ± 2.21[a] | 0.017 |
| Mean corpuscular volume (fL) | 53 ± 0[b] | 47 ± 2[a] | 52 ± 1[ab] | 49 ± 2[ab] | 0.028 |
| Mean corpuscular hemoglobin concentration (g/dL) | 29.7 ± 0.5[a] | 33.8 ± 0.2[b] | 30.0 ± 0.2[a] | 32.1 ± 0.3[ab] | 0.012 |
| White blood cells ($10^3$/mm$^3$) | 3.58 ± 0.40 | 3.75 ± 0.39 | 3.85 ± 0.46 | 2.85 ± 0.19 | 0.257 |
| Band neutrophils (%) | 0 ± 0 | 0 ± 0 | 0 ± 0 | 0 ± 0 | 0.650 |
| Segmented neutrophils (%) | 34 ± 3 | 34 ± 4 | 34 ± 4 | 29 ± 6 | 0.825 |
| Eosinophils (%) | 1 ± 1 | 0 ± 0 | 0 ± 0 | 1 ± 1 | 0.424 |
| Lymphocytes (%) | 65 ± 2 | 65 ± 4 | 65 ± 4 | 69 ± 5 | 0.834 |
| Monocytes (%) | 1 ± 0 | 1 ± 1 | 1 ± 0 | 0 ± 0 | 0.813 |
| Platelets ($10^3$/mm$^3$) | 704 ± 47 | 640 ± 44 | 701 ± 56 | 626 ± 136 | 0.858 |

Note:
Data are expressed as mean ± standard error of the mean. Statistically significant effect: values of one parameter without common superscript (a,b) are statistically significantly different at a significance level of $p < 0.05$. $n = 6$. KYNA, kynurenic acid.

**Table 6  Biochemical blood serum parameters.**

| Parameter | Experimental diet | | | | p |
|---|---|---|---|---|---|
| | Control | KYNA | Low-KYNA yeast | High-KYNA yeast | |
| Total protein (g/L) | 56.1 ± 1.0[B] | 47.7 ± 2.0[A] | 55.8 ± 1.3[B] | 50.7 ± 0.9[AB] | 0.003 |
| Albumins (g/L) | 46.9 ± 2.9 | 43.9 ± 1.7 | 48.0 ± 1.1 | 43.9 ± 1.2 | 0.143 |
| Globulins (g/L) | 9.2 ± 2.0 | 3.9 ± 1.1 | 7.7 ± 1.4 | 6.8 ± 0.8 | 0.056 |
| Albumin/globulin ratio | 5.7 ± 1.4 | 11.0 ± 2.0 | 8.2 ± 2.5 | 7.3 ± 1.4 | 0.422 |
| Total cholesterol (TC) (mmol/L) | 1.04 ± 0.06[A] | 1.13 ± 0.03[AB] | 1.33 ± 0.04[B] | 1.30 ± 0.07[B] | 0.002 |
| High-density lipoproteins (HDL) (mmol/L) | 0.57 ± 0.06 | 0.46 ± 0.05 | 0.63 ± 0.05 | 0.52 ± 0.05 | 0.192 |
| TC/HDL ratio | 1.96 ± 0.24 | 2.54 ± 0.20 | 2.19 ± 0.17 | 2.57 ± 0.13 | 0.098 |
| Triacylglycerols (mmol/L) | 2.82 ± 0.36 | 2.88 ± 0.34 | 3.26 ± 0.44 | 3.06 ± 0.12 | 0.785 |
| Glucose (mmol/L) | 1.54 ± 0.15 | 1.64 ± 0.17 | 1.69 ± 0.14 | 1.48 ± 0.16 | 0.780 |
| Aspartate aminotransferase (U/L) | 219.1 ± 69.1 | 296.8 ± 70.0 | 273.2 ± 47.1 | 275.5 ± 55.1 | 0.843 |
| Alanine aminotransferase (U/L) | 28.6 ± 2.8 | 31.0 ± 4.3 | 29.7 ± 4.6 | 33.7 ± 4.5 | 0.848 |
| Lactate dehydrogenase (U/L) | 535.0 ± 105.9 | 829.7 ± 154.2 | 812.1 ± 140.8 | 975.9 ± 160.2 | 0.404 |
| Gamma-glutamyl transpeptidase (U/L) | 1.50 ± 0.40 | 3.30 ± 1.46 | 3.52 ± 0.55 | 1.97 ± 0.56 | 0.450 |
| Urea (mmol/L) | 0.35 ± 0.02[A] | 0.51 ± 0.02[AB] | 0.48 ± 0.04[AB] | 0.59 ± 0.04[B] | 0.006 |
| Creatinine (μmol/L) | 2.88 ± 0.40 | 3.79 ± 0.63 | 4.83 ± 0.77 | 5.01 ± 1.18 | 0.535 |
| Urea/creatinine ratio | 133.24 ± 13.58 | 151.40 ± 24.79 | 111.62 ± 21.43 | 146.85 ± 27.46 | 0.654 |

Note:
Data are expressed as mean ± standard error of the mean. Statistically significant effect: values of one parameter without common superscript (A,B) are statistically significantly different at a significance level of $p < 0.01$. KYNA, kynurenic acid.

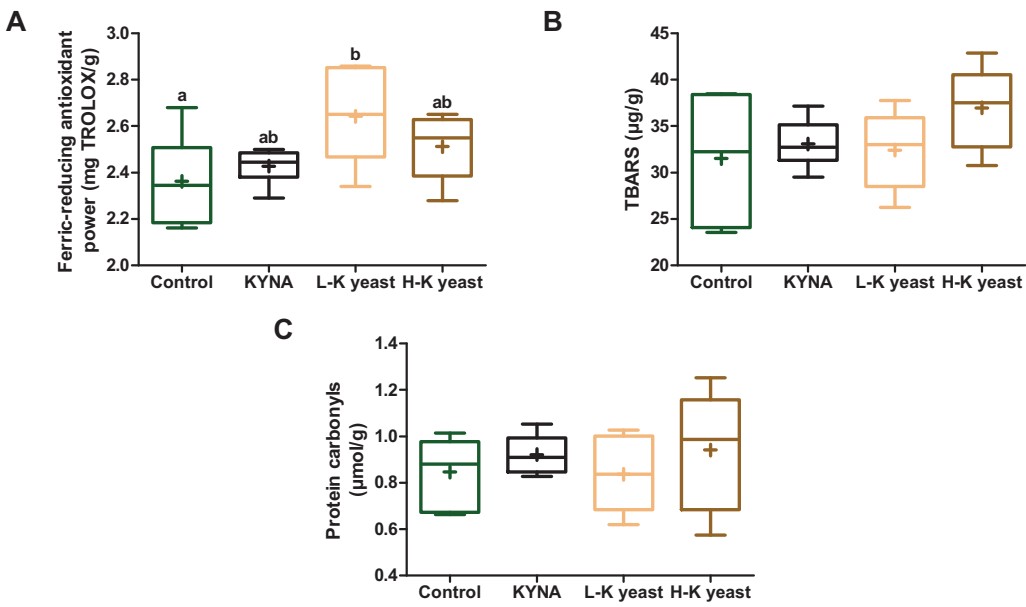

**Figure 3** **Box and whiskers plots for redox state indicators.** (A) Ferric-reducing antioxidant power; (B) TBARS (thiobarbituric acid reactive substances) and (C) protein carbonyls in the livers of mice. Boxes enclose the interquartile range, whiskers—min to max values, horizontal lines—the median, "+"—the mean. Statistically significant effect: values of one parameter without common letter (a,b) are statistically significantly different at a significance level of $p < 0.05$. $n = 6$. Control, control diet; KYNA, kynurenic acid diet; L-K yeast, low-KYNA yeast diet; H-K yeast, high-KYNA yeast diet.

noting that the cumulative body weight gain per cage of mice fed the low-KYNA yeast diet was significantly greater than that of the animals fed the control diet and PER tended to increase in case of this diet (Figs. 2A, 2C).

## Hematological and biochemical blood parameters

Mice ingesting the high-KYNA yeast diet tended to have lower hematocrit and mean corpuscular volume and higher mean corpuscular hemoglobin concentration in the blood than mice fed the control diet (Table 5). In turn, mice consuming the KYNA diet had lower mean corpuscular volume and higher mean corpuscular hemoglobin concentration. Moreover, mice ingesting the high-KYNA yeast diet tended to have lower serum total protein and globulins compared to mice receiving the control diet (Table 6). Mice fed the KYNA diet showed a lower total protein concentration and a trend towards lower globulins. Furthermore, animals ingesting diets with two types of yeast had significantly higher serum TC level than mice fed the control diet, with no significant differences in TC/HDL between the groups. In turn, mice on a high-KYNA yeast diet had significantly greater serum urea level than mice on the control diet. There were no significant differences in the influence of the experimental diets on the remaining blood hematological and biochemical markers (Tables 5 and 6).

## Liver redox state indicators

Intake of high-KYNA yeast and KYNA diets resulted in a trend towards greater ferric-reducing antioxidant power in the liver compared to the liver of mice receiving the control diet, and consumption of low-KYNA yeast diet increased this indice (Fig. 3A). There was no statistically significant influence of the administration of the experimental diets on the TBARS and protein carbonyls (Figs. 3B, 3C).

## DISCUSSION

KYNA content in foods ranges from trace amounts to a highest reported concentration of about 0.6 g/kg in chestnut honey (Turska et al., 2022). Our previous study shows that the most effective method of KYNA production in *Y. lipolytica* yeast strain S12 is the use of 40 g/L of fructose and 200 mg/L of Trp, which is a precursor of KYNA synthesis, in the medium or the addition of above honey (Wróbel-Kwiatkowska et al., 2020a, 2020b). For economic reasons, fructose and Trp were selected for high-KYNA yeast production in this study. KYNA concentration in this dry yeast biomass was 0.80 + 0.08 g/kg, while in low-KYNA yeast, obtained without the addition of Trp - 0.29 + 0.01 g/kg.

It turned out that the high-KYNA yeast was characterized by a lower content of crude protein and all amino acids than low-KYNA yeast. Crude protein concentration in high-KYNA yeast was lower than in *Saccharomyces cerevisiae* brewer's yeast in studies by *Czech et al. (2016, 2018a)*. Moreover, the crude protein of the high-KYNA yeast had a lower content of most individual amino acids compared to the protein of the low-KYNA yeast, with the exception of higher Ser content and unchanged Thr, Gly and Pro content. Thus, it can be assumed that providing conditions for increased KYNA production affects amino acid metabolism in yeast. However, the concentration of Ser, Thr, Gly and Ala was greater than in *S. cerevisiae* protein (Czech et al., 2016, 2018a). The high-KYNA yeast also contained less minerals than the low-KYNA yeast and their concentration was lower than in *S. cerevisiae* (Czech et al., 2016, 2018a). Other authors confirmed the presence of numerous macro- and microelements in *Y. lipolytica* (Jach & Malm, 2022; Merska, Czech & Ognik, 2015; Czech et al., 2016, 2018a).

On the other hand, high-KYNA yeast was characterized by a higher concentration of crude fat and more than twice as much crude fiber content as the low-KYNA yeast, and the concentration of these components was much greater than in the case of *S. cerevisiae* (Czech et al., 2016, 2018a).

Antioxidants prevent the production of reactive oxygen species (ROS), which high concentration in cells leads to oxidative stress, resulting in damage to lipids, proteins and DNA, contributing to many diseases, and inhibit the activity of ROS (Matusiewicz et al., 2022). The high-KYNA yeast was characterized by a lower antioxidant potential expressed as the ferric-reducing antioxidant power, based on the reduction of $Fe^{3+}$ to $Fe^{2+}$, than the low-KYNA yeast. The lower concentration of phenols and GSH contributed to the lower antioxidant potential of this yeast. GSH is synthesized from glutamate, cysteine and glycine, therefore the lower protein content in high-KYNA yeast may have contributed to the lower GSH content (Łukaszewicz-Hussain, 2003). It is involved in protecting cells from damage caused by ROS, reactive nitrogen species, lipid peroxides and xenobiotics, and

controls many important cellular processes and immune functions (*Matusiewicz et al., 2022*). The lower antioxidant potential of high-KYNA yeast could have also been influenced by the lower concentration of methionine and minerals (*Czech, Merska-Kazanowska & Całyniuk, 2020*; *Matusiewicz et al., 2022*). Mannan oligosaccharides and β-glucans of the yeast cell wall also contributed to the antioxidant potential of yeast (*Czech, Merska-Kazanowska & Całyniuk, 2020*), and in the case of our experiment—especially high-KYNA yeast. In turn, *Reyes-Becerril, Alamillo & Angulo (2021)* confirmed the antioxidant activity of two *Y. lipolytica* strains. In addition, greater concentration of KYNA contributed to the antioxidant potential of high-KYNA yeast. The antioxidant properties of this compound were confirmed among others by the reduction of iron ions (*Li et al., 2021*). KYNA's antioxidant potential was also demonstrated by *Dhakar et al. (2019)* and it may be due to the aromatic hydroxyl group which easily provides protons for reaction with free radicals. In addition, the research by *Lugo-Huitrón et al. (2011)* showed that KYNA is capable of scavenging hydroxyl radical ($\cdot OH$), superoxide anion ($O_2 \cdot^-$) and peroxynitrite anion ($ONOO^-$), and its $O_2 \cdot^-$ scavenging capacity is one order of magnitude greater than that of GSH. In turn, *Pérez-González, Alvarez-Idaboy & Galano (2015)* predicted that KYNA is one of the best free radical scavengers among the tested tryptophan metabolites, 24 times more effective in scavenging the hydroperoxyl radical ($\cdot OOH$) than TROLOX (6-hydroxy-2,5,7,8-tetramethylchroman-2-carboxylic acid), considering only SET (single electron transfer) processes. Moreover, *Genestet et al. (2014)* showed that KYNA has a better ability to scavenge $O_2 \cdot^-$ than other tryptophan metabolites, and it also scavenges $H_2O_2$. In addition, a possible mechanism for the oxidation of KYNA by $\cdot OH$, sulfate radical anion ($SO_4 \cdot^-$) and oxide radical anion ($O \cdot^-$) is described by *Prasanthkumar, Sajith & Singh (2020)*. In the present study, a greater concentration of KYNA in high-KYNA yeast, with a considerably changed chemical composition, was insufficient to increase its antioxidant potential relative to low-KYNA yeast.

In our 7-week research, the cumulative body weight gain per cage of mice fed the high-KYNA yeast diet was not different from mice fed the control diet, with no changes in cumulative food intake per cage and PER, indicating the nutritional quality of the protein, between these groups. In turn, in the experiment of *Michalik et al. (2013)*, feeding young Wistar rats diets in which 40% of cereal protein was replaced with *Y. lipolytica* protein resulted in higher PER. In our study, we observed a trend towards higher PER in the case of the mice fed the low-KYNA yeast diet, compared to mice receiving the control diet, which was associated with greater cumulative body weight gain per cage. The lack of effect of the high-KYNA yeast diet on cumulative body weight gain per cage may have been due to the greater content of good quality fiber, among others stimulating the immune activity and modulating the intestinal microflora, in the high-KYNA yeast than in the low-KYNA yeast (*Czech et al., 2018b*). In turn, in a 39-day research, a 3% addition of *Y. lipolytica* to piglet feed increased body weight gain and decreased feed conversion ratio (FCR) without affecting feed intake (*Czech et al., 2016*). In other 8-week experiment on piglets, a 3% addition of *Y. lipolytica* also resulted in a greater body weight gain (*Czech et al., 2018a*). Contrary to our results, its addition reduced feed intake and FCR. We did not note the
effect of KYNA on body weight gain, however in the study of *Milart et al. (2019)*, administration of KYNA in water (250 mg/L) to mothers of young male Wistar rats, from 1 to 21 days of age, resulted in a decrease in body weight of the young animals. In another 40-day experiment, the administration of KYNA in water (25 or 250 mg/L) to 21-day-old male and female Wistar rats also resulted in a reduction in body weight (*Tomaszewska et al., 2019*). These experiments indicate the anti-obesity properties of KYNA over early development. In turn, in the study by *Agudelo et al. (2018)*, body weight of adult C57BL/6J mice, on a high-fat or control diet, was limited due to the activation of GPR35, dependent on KYNA, which was administered intraperitoneally (5 mg/kg/day). Higher energy expenditure, better energy metabolism and positive effect on inflammation, mainly in adipose tissue, were confirmed. Moreover, KYNA, as an agonist of GPR35 receptors, may be part of the gut-brain signal axis responsible for the regulation of energy balance (*Quon et al., 2020*). In another study, intragastric administration of KYNA (5 mg/kg/day) to 4-week-old Kunming mice on a high-fat diet, for 8 weeks, reduced the body weight gain and food intake (*Li et al., 2021*). The authors attribute the beneficial effect of KYNA to its antioxidant properties and modulation of the gut microbiota. On the other hand, administration of KYNA in water (250 mg/L), for 21 days, did not influence the body weight of old female Balb/c mice and adult male Wistar rats (*Turski et al., 2014*). In our experiment, the lack of negative effect of KYNA on body weight gain may have been related to the fact that the animals were adults and received diets with standard levels of fat and energy.

Mice consuming the high-KYNA yeast diet tended to have lower hematocrit and mean corpuscular volume than mice receiving the control diet which may have been the result of a trend towards decrease in total protein concentration in the blood serum. The tendency to increase in mean corpuscular hemoglobin concentration may be considered as compensation for a decrease in mean corpuscular volume. Furthermore, it is known from the literature that smaller erythrocytes have a greater ratio of surface area to volume and a shorter diffusion distance, which translates into faster hemoglobin oxygenation and deoxygenation (*Penman, Deeming & Soulsbury, 2022*). However, the shape of the erythrocytes varies plastically within a given species. Moreover, smaller erythrocyte size may also be associated with a higher metabolic rate, which we did not notice in our experiment. However, in the research of *Turski et al. (2014)*, administration of KYNA in water (25 or 250 mg/L), for 3 and 14 days, did not influence the hematocrit. In other 8-week experiment, a 3% addition of *Y. lipolytica* to piglet feed resulted in an increase in red blood cells, hemoglobin, hematocrit, white blood cells and lymphocytes, as well as a reduction in neutrophils, which we did not observe in our study (*Czech et al., 2018b*). In another 15-week experiment, 3% and 6% addition of *Y. lipolytica* increased the proportion of monocytes in the blood of turkey hens (*Czech, Merska & Ognik, 2014*).

Mice consuming the high-KYNA yeast diet tended to have lower serum total protein and globulins than mice receiving the control diet. In turn, in fish, KYNA (250 mg/kg feed), administered for 7 days, demonstrated a temporary immunomodulatory effect—it inhibited the proliferation of B lymphocytes, indirectly responsible to produce one of the globulin fractions—immunoglobulins (*Małaczewska et al., 2013*). Contrary to our results,

in other experiment of *Monfared et al. (2023)*, on BALB/c mice orally administered KYNA (1.1 mg/mouse/day), for 1 week, no changes in total protein content in blood plasma were observed. In the present study, mice fed the yeast diets showed higher serum TC than animals on the control diet, with no differences in TC/HDL between the groups. According to *Li et al. (2021)*, blood serum cholesterol can be transported to the liver by HDL and excreted. Moreover, we noted that mice on a high-KYNA yeast diet had greater concentration of serum urea and this indicator can be influenced by many extrarenal factors (*Lyman, 1986*). In turn, the 3% and 6% addition of *Y. lipolytica* increased the concentration of urea, creatinine and LDH and decreased the content of TC and triacylglycerols in the blood plasma of turkey hens; the addition of 3% reduced the concentration of ALT; and 6% increased total protein, % HDL, ALT, reduced AST (*Czech, Merska & Ognik, 2014*). In another study of *Czech, Merska-Kazanowska & Całyniuk (2020)*, a 3% addition of *Y. lipolytica* to the feed increased total protein and creatinine and decreased TC, HDL, triacylglycerols, as well as AST and ALT in turkey hen blood plasma. In turn, in the experiment of *Czech et al. (2018a)*, the addition of *Y. lipolytica* caused a decrease in TC and an increase in HDL and glucose in the blood plasma of piglets. In addition, in the experiment of *Monfared et al. (2023)*, in mice administered KYNA, a slight increase in the concentration of urea in blood plasma and higher AST and ALT values were observed. Moreover, in the experiment of *Li et al. (2021)*, administration of KYNA to mice on a high-fat diet decreased the level of triacylglycerols and increased the level of HDL in the blood serum.

In our study, ingestion of high-KYNA yeast diet resulted in a trend towards higher ferric-reducing antioxidant power in the liver compared to mice consuming the control diet, however intake of low-KYNA yeast diet increased this indice. Phenols, GSH, cysteine and other antioxidants contained in yeast, especially low-KYNA yeast, could have had a positive effect on the redox status in the liver. Moreover, the ash content, lower in the high-KYNA yeast than in the low-KYNA yeast, could have affected the activity of antioxidant enzymes. The increased concentration of serum urea, a low-molecular antioxidant, was, in turn, the result of the consumption of high-KYNA yeast diet. In turn, in the research of other authors, the addition of *Y. lipolytica* resulted in a tendency towards increase in the total antioxidant potential, based on the mechanism of $Fe^{3+}$ reduction to $Fe^{2+}$, and also towards increase in the activity of SOD and catalase, increased the content of vitamin C and decreased the concentration of the lipid peroxidation end product, malondialdehyde (MDA) in the blood plasma of turkey hens (*Czech, Merska-Kazanowska & Całyniuk, 2020*). Higher plasma zinc and iron levels may have contributed to higher SOD and catalase activities. In addition, in another 15-week experiment, 3% and 6% addition of *Y. lipolytica* to the feed resulted in a trend towards increase in the total antioxidant potential ($Fe^{3+}$ reduction to $Fe^{2+}$) in turkey hen blood plasma, and also increased the activity of catalase, which was accompanied by a higher concentration of iron, and decreased MDA (*Merska, Czech & Ognik, 2015*).

In our study, higher content of KYNA may have greatly contributed to the ferric-reducing antioxidant power in the livers of the mice that ingested high-KYNA yeast diet. In turn, in the experiment of *Monfared et al. (2023)*, mice treated with KYNA had

higher total antioxidant status, SOD activity and glutathione peroxidase levels, and lower MDA concentrations. In other research, all doses of KYNA (2.5, 25, 250 mg/L), administered to mice in water, reduced the oxidative burst in granulocytes and monocytes, which was probably related to its antioxidant properties (*Małaczewska et al., 2014*). In turn, according to *Bratek-Gerej et al. (2021)*, administration of KYNA intraperitoneally (50, 150, 300 mg/kg) to a rat model of neonatal hypoxia-ischemia resulted in a reduction in ROS level and activity of antioxidant enzymes (SOD, glutathione peroxidase and catalase) and a partial restoration of GSH content in the brain. Additionally, injection of KYNA into the striatum of 30-day-old male Wistar rats did not affect ROS content, SOD/catalase ratio, glutathione peroxidase activity and sulfhydryl content in this structure, however, KYNA counteracted changes in redox homeostasis induced by quinolinic acid (*Ferreira et al., 2020*). In turn, KYNA administered to female albino rats by oral gavage (2.5 mg/kg/day), for 14 days, counteracted monosodium glutamate—induced ovarian GSH decrease and MDA increase (*Elraey et al., 2021*). In our study, the high-KYNA yeast diet did not affect the degree of oxidation of lipids and proteins in the liver of mice, which can be explained by the lack of a nutritional or environmental factor to induce them.

## CONCLUSIONS

The improvement of the quality of *Y. lipolytica* yeast biomass with increased content of KYNA, including its antioxidant potential, would be affected by the preserved level of protein and unchanged amino acid profile. It will be worth investigating the effect of such optimized yeast on model animals, including animals with metabolic diseases. It is also essential to study the influence of this yeast on other indicators of health. It may be of interest to consumers as a novel food or dietary supplement.

## ACKNOWLEDGEMENTS

This work is part of a habilitation thesis by Magdalena Matusiewicz.

### Funding

The authors received no funding for this work.

### Competing Interests

The authors declare that they have no competing interests.

### Author Contributions

- Magdalena Matusiewicz conceived and designed the experiments, performed the experiments, analyzed the data, prepared figures and/or tables, authored or reviewed drafts of the article, and approved the final draft.
- Magdalena Wróbel-Kwiatkowska conceived and designed the experiments, performed the experiments, analyzed the data, authored or reviewed drafts of the article, and approved the final draft.

- Tomasz Niemiec conceived and designed the experiments, performed the experiments, analyzed the data, authored or reviewed drafts of the article, and approved the final draft.
- Wiesław Świderek conceived and designed the experiments, performed the experiments, analyzed the data, authored or reviewed drafts of the article, and approved the final draft.
- Iwona Kosieradzka conceived and designed the experiments, analyzed the data, authored or reviewed drafts of the article, and approved the final draft.
- Aleksandra Rosińska performed the experiments, authored or reviewed drafts of the article, and approved the final draft.
- Anna Niwińska performed the experiments, authored or reviewed drafts of the article, and approved the final draft.
- Magdalena Rakicka-Pustułka performed the experiments, authored or reviewed drafts of the article, and approved the final draft.
- Tomasz Kocki performed the experiments, authored or reviewed drafts of the article, and approved the final draft.
- Waldemar Rymowicz conceived and designed the experiments, performed the experiments, analyzed the data, authored or reviewed drafts of the article, and approved the final draft.
- Waldemar A. Turski conceived and designed the experiments, analyzed the data, authored or reviewed drafts of the article, and approved the final draft.

### Data Availability

The raw data are available in the Supplemental File.

### Supplemental Information

Supplemental information for this article can be found online at http://dx.doi.org/10.7717/peerj.15833#supplemental-information.

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
