# Peer review of "Effect of Yarrowia lipolytica yeast biomass with increased kynurenic acid content on selected metabolic indicators in mice"

_PeerJ, doi:10.7717/peerj.15833_

## Round 0.1 · original submission · Major Revisions

Please address the concerns of both reviewers and amend the manuscript accordingly.

·

Basic reporting

The imanyscript is written in a very unacceptable way.
Simply authors put all what they get from experiments and what they read about.
Results need to be critically reviewed and selected towards clear aim of the study. Terribly long manuscript (especially discussion) must be abbreviated by 2/3.
80-thy references are almost enough for review article!
Being not native english speaker I can say only that I can understand scientific message of the Manuscript

Experimental design

Results and design need to be critically reviewed and corrected towards clear aim of the study.

Validity of the findings

Information you present in the manuscript is interesting.
However, you put in the manuscript content all what you know. Suggestions is to shorten the manuscript by 2/3 part. Clearly say what was the aim of the study. 80-thy references are good for specific review. The length of discussion is unbelieved.

Let leave in the manuscript content the most important parts helping you to formulate your own Conclusions.

Additional comments

Some -technical suggestions:

Set up clear aim of the study.

Tables and Figures
-Tables 2 and 5. Delete the rows concerning aromatic amino acids (AAA) and AAA/TAA ratio, and # in the superscript of glutamic acid, aspartic acid, alanine, and glycine. These amino acids are not relevant to the article.
-Figure 1. Delete Figure 1B and C. Differences for these indicators were statistically insignificant.
-Table 4. Delete diet components from” Others”. In the Materials and Methods section, there is a citation of an article (Reeves, 1997) where these components are listed.
-Delete Table 6. This table does not contribute important information to the article; Table 5 is sufficient.
-Delete Figure 2C. The data in this Figure are the final points in Figure 2B.
-Delete Figure 3. This Figure does not contribute important information to the article.
-Delete Figure 4B, C, D. Differences for these indicators were statistically insignificant.
-Table 9. Delete the rows concerning the concentration of KYNA, KYN and Trp in the cecum. These contents in the cecum do not contribute important information to the article.
Also, delete the above information from all sections of the article.
-Abstract - adjust to be consistent with the content of the corrected article.
-Introduction - delete a fragment from lines 131-136. This fragment is unnecessary because, in the PeerJ journal, the Materials & Methods section is before the Results section.
Materials & Methods
-Media and Batch culture sections (lines 145-165) - quote repetitive information from an earlier article (Wróbel-Kwiatkowska et al., 2020a).
-Analysis of amino acids in yeast biomass section (lines 197-200) - quote repetitive information from an earlier article (Wróbel-Kwiatkowska et al., 2020b).
-Determination of yeast biomass antioxidant indicators section (lines 213-217) - delete the part of the methodology regarding Figures 1B and C (to be deleted). Include in this section the following methods for determining phenols and glutathione in yeast biomass (lines 218-223).
-Lines 262-265 - correct this part of the methodology, considering the removal of Figure 3 and the lack of presentation of the results of concentrations of KYNA, KYN and Trp in cecum in the revised version of the article (Table 9).
-Lines 268 - delete information on AAS, CS and EAAI (deletion of Table 6).
-Lines 298-301 - correct this part of the methodology, considering the removal of Figures 4B, C, and D.
-Determination of KYNA, kynurenine (KYN) and Trp in the liver, spleen, cecum, and kidney and KTR calculation for the liver and spleen section (lines 329-345) - delete information that is repeated with details in the Determination of kynurenic acid (KYNA) in yeast biomass section and refer to this section. Delete "cecum" from this section.
Results
-Lines 384-386 - delete information concerning aromatic amino acids
-Lines 388-392 - shorten the section's title to "Antioxidant indicators". Delete the results for Figures 1, B, and C.
-Lines 402-404 - delete the results for Table 6.
-Lines 406-412 - "cumulative" instead of "total."
-„Weight of selected internal organs” section (lines 434-438) – delete
-Lines 443-444 - delete information concerning Figure 4B, C, D.
-Lines 449-452 - remove contents in the cecum and modify.
Discussion
-Delete information regarding data removed from Tables and deleted Figures
-Significantly shorten the Discussion, leaving the information most relevant to the hypothesis and most leading to Conclusions. Limit your citations significantly. Connect some sentences together.
Conclusions
Prepare shorter Conclusions, focusing on highlighting the scientific value of the work.

Reviewer 2 ·

Basic reporting

no comments

Experimental design

no comments

Validity of the findings

no comments

Additional comments

Referees Comments

on the manuscript entitled " Effect of Yarrowia lipolytica yeast biomass with increased kynurenic acid content on selected metabolic indicators in mice” for Peer J

The manuscript by M. Matusiewicz and co-workers (Effect of Yarrowia lipolytica yeast biomass with increased kynurenic acid content on selected metabolic indicators in mice) is certainly within the scope of Peer J. In general, the manuscript is well written, but there are several points that should be addressed before the manuscript is considered for publication.
Point 1: Line 152: Please to check up “1 g/L yeast”.
Point 2: Lines 150-155: Please to include the composition of seed medium.
Point 3: Lines 364 – 368: Please correct the text to the data provided in Table 2.
Point 4: Table 3 – Please to include the data of amino acid and EAAI of FAO/Who standard protein.
Point 5: Lines 381-387 – Move the description of the data presented in Table 2 above the text (before describing the data from Table 3).
Point 6: Line 384: Among the amino acids, the following are classically considered aromatic: phenylalanine, tryptophan and tyrosine.
Point 7: Lines 394-397 and Table 4: Please to include a description of the data presented in Table 4 or delete this part.

---

## Round 0.2 · accepted · Accept

In my view, all the issues pointed by the reviewers were adequately addressed and the manuscript was amended accordingly. Therefore, revised version is acceptable in its present form.